# Feedback Beamforming in the Time Domain

**DOI:** 10.3390/s24072179

**Published:** 2024-03-28

**Authors:** Zvi Aharon Herscovici, Israel Cohen

**Affiliations:** Andrew and Erna Viterbi Faculty of Electrical & Computer Engineering, Technion—Israel Institute of Technology, Haifa 3200003, Israel; icohen@ee.technion.ac.il

**Keywords:** beamforming, time domain processing, source localization, uniform linear array

## Abstract

Real-time source localization is crucial for high-end automation and artificial intelligence (AI) products. However, a low signal-to-noise ratio (SNR) and limited processing time can reduce localization accuracy. This work proposes a new architecture for a time-domain feedback-based beamformer that meets real-time processing demands. The main objective of this design is to locate reflective sources by estimating their direction of arrival (DOA) and signal range. Incorporating a feedback mechanism in this architecture refines localization precision, a unique aspect of this approach. We conducted an in-depth analysis to compare the effectiveness of time-domain feedback beamforming against conventional time-domain methods, highlighting their benefits and limitations. Our evaluation of the proposed architecture, based on critical performance indicators such as peak-to-sidelobe ratio, mainlobe width, and directivity factor, demonstrates its ability to improve beamformer effectiveness significantly.

## 1. Introduction

Array signal processing has profound applications across radar and sonar systems [1,2,3], smart antennas for satellite and cellular communications [4,5,6,7,8], automotive radar [9,10], the early detection of diseases using medical imaging [11,12], and recently, reconfigurable intelligent surface (RIS) applications [7,13,14]. This work delves into the critical task of source localization in noisy environments, employing sensor arrays to pinpoint a signal’s origin. Source localization is an essential aspect of spatial signal processing. It involves using sensor arrays to detect and determine the origin of signals in different environments. This process starts with a reference sensor emitting signals into space while the sensor array captures reflections from various sources. By analyzing the time delays between the readings of these sensors, it is possible to estimate the exact location of the source accurately.

Beamforming is a technique used in spatial processing that helps detect a signal coming from a specific direction while minimizing the influence of noise and interference from other directions [15]. This technique uses an array of sensors and determines the direction of the signal by weighing the inputs from each sensor. The goal is to enhance the signal from the target direction while reducing interferences. When beamforming is implemented on digital platforms, it requires the discretization of the signal, which introduces quantization errors and necessitates a higher sampling rate. Because of these factors, research has focused on frequency domain approaches [14,16,17], which require lower sampling rates and can benefit from the combined utilization of spectral and spatial data. However, the time domain is still an important area for development [3,7,12,18,19], particularly for applications that require low latency [19], such as real-time communication [3,12]. In such cases, the time-domain approach can reduce computational complexity and execution time, even with a limited number of sensors.

Sensor arrays are available in different configurations, such as uniform linear [20,21,22], circular [23,24,25], and planar arrays [26,27], each having its benefits and challenges. Uniform linear arrays (ULAs) have been a significant area of research due to their simple implementation and easy analysis. The design of a ULA, including the number of elements and their spacing, significantly impacts its performance, affecting the sharpness of the mainlobe and sidelobe levels. Choosing the right weight is crucial for optimizing the beamformer’s performance. Recent advancements have introduced a unique frequency domain feedback beamforming architecture [17], which incorporates a feedback loop to enhance source localization and signal rebroadcasting. Although this approach introduces additional complexity, it offers better system tuning by integrating an infinite-impulse response (IIR) filter. This circumvents the challenges associated with temporal processing while maintaining a low-complexity array.

This paper proposes a new architecture for a time-domain feedback beamformer to meet real-time processing demands. This architecture aims to locate a reflective source by estimating its direction of arrival (DOA) and range while incorporating a feedback mechanism to improve accuracy. In contrast to conventional array processing and typical beamforming methods, the novelty lies in integrating spatial feedback, resembling temporal domain IIR filtering in the spatial domain. Assuming the target of interest exhibits reflective characteristics akin to a mirror, the spatial feedback between the array and the target is established by retransmitting a synthesized version of the incoming signal back to the target. We have compared this new architecture against established metrics such as peak-to-sidelobe ratio, mainlobe width, and directivity factor, demonstrating its potential to improve the beamformer performance significantly. The contributions of this manuscript are twofold:

(1) We have developed a new beamforming architecture that includes feedback mechanisms. This innovation offers a comprehensive methodology for integrating feedback mechanisms into beamforming systems. We have presented a theoretical framework and a closed-form solution for a time-domain feedback-based beamformer. This new architecture extends the current beamforming applications and provides an efficient way of implementing feedback mechanisms in beamforming systems.

(2) We have conducted an extensive analysis comparing feedback beamforming in the time domain with conventional time-domain methodologies, highlighting their operational strengths and limitations. We aim to present each approach’s distinctive features and their respective efficacies. Our empirical results emphasize the effectiveness of the proposed beamforming framework in improving target localization estimation, thus showcasing its potential usefulness in various real-world scenarios.

The paper is structured as follows: We start in Section 2 by explaining the signal model for impinging signals on a ULA in the time domain. Next, in Section 3, we delve into the design of the feedback beamformer and elaborate on the signal model within this innovative architecture. In Section 4, we propose a methodology to optimize beamformer weights for the precise estimation of the direction of arrival and signal range. In Section 5, we present simulation results, demonstrating significant improvements in spatial performance metrics compared to conventional beamformers. In Section 6, we compare the performances of time-domain and frequency-domain implementations. Finally, in Section 7, we conclude with a discussion of the implications of our findings and suggest future research directions.

Notations: Variables are represented in italics, and matrices and vectors are distinguished by boldfaced type, with matrices in uppercase and vectors in lowercase. The superscript ^*T*^ denotes the transpose operation for matrices or vectors. The elements within vectors and matrices are referenced as follows: $v_{i}$ indicates the *i*th element within vector v, and $A_{ij}$ specifies the element located at the *i*th row and *j*th column of matrix A.

## 2. Signal Model

Consider a far-field source signal, s(t), that propagates in an anechoic environment at the speed of light, *c*. The signal s(t), which is an impulse or continuous signal, impinges an object and reflects the sensor array. A ULA beamformer, functioning analogously to an FIR filter in the time domain, is utilized in the design. The ULA consists of *N* omnidirectional elements with inter-element spacing, δ. The location of the *n*th sensor is denoted by $p_{n}$ for n=0,…,N−1, with $p_{0}$ representing the reference point relative to a stationary target positioned at $p_{t}$. For simplicity, we consider a two-dimensional formulation. The DOA of the reflected signal, relative to the broadside direction of the array, is denoted as $\theta_{d}$. The distance from the reference sensor to the target is given by $d=\left|p_{0}-p_{t}\right|$. The configuration of ULA is depicted in Figure 1.

The signal measured at the *n*th sensor is given by
(1)$$x_{n}\left(t\right)=g_{c}\;s\left(t-\tau_{\mathrm{pd}}-\tau_{n}\right)+v_{n}\left(t\right)\;,$$
where $g_{c}$ is the channel gain, $\tau_{\mathrm{pd}}$ is the prorogation delay from the target to the reference sensor and back, given by $\tau_{\mathrm{pd}}=\frac{2d}{c}$, $\tau_{n}$ is the time delay between the *n*th and reference sensors, which is given by $\tau_{n}=\frac{n\delta \cos(\theta_{d})}{c}$, and $v_{n}\left(t\right)$ is the noise in the *n*th sensor.

Following Shannon’s sampling theorem [28], the continuous-time signal can be represented as
(2)$$s\left(t\right)=\sum_{m=-\infty }^{\infty }s\left[m\right]\;\mathrm{sinc}\left(tf_{s}-m\right)$$
where $f_{s}=\frac{1}{T_{s}}$ is the sampling frequency, $T_{s}$ is the sampling interval, $s\left[m\right]=s\left(mT_{s}\right)$ is the sampled signal, m∈Z is the discrete-time index, and $\mathrm{sinc}\left(x\right)=\frac{\sin(\pi x)}{\pi x}$. Substituting (Equation 1) into (Equation 2), and under the assumption of a noiseless environment, where $v_{n}\left(t\right)=0$ for all the sensors, we have
(3)$$x_{n}\left(t\right)=g_{c}\;\sum_{m=-\infty }^{\infty }s\left[m\right]\;\mathrm{sinc}\left(\left(t-\tau_{\mathrm{pd}}-\tau_{n}\right)f_{s}-m\right).$$

The discrete-time signal measured in the *n*th sensor can be written as
(4)$$\begin{array}{cc} x_{n}\left[m\right] & =x_{n}\left(mT_{s}\right)\\ \; & =g_{c}\;\sum_{m^{\prime }=-\infty }^{\infty }s\left[m^{\prime }\right]\;\mathrm{sinc}\left(\left(mT_{s}-\tau_{\mathrm{pd}}-\tau_{n}\right)f_{s}-m^{\prime }\right)\\ \; & =g_{c}\;\sum_{l=-\infty }^{\infty }s\left[m-l\right]\;\mathrm{sinc}\left(l-\left(\tau_{\mathrm{pd}}+\tau_{n}\right)f_{s}\right)\;. \end{array}$$

By following the framework presented in [29] for general analysis and by defining $\Delta =\tau_{\mathrm{pd}}f_{s}$, we have
(5)$$\begin{array}{cc} x_{n}\left[m\right] & =g_{c}\sum_{l=-\infty }^{\infty }s\left[m-l\right]\;\mathrm{sinc}\left(l-\Delta -f_{s}\tau_{n}\right)\\ \; & \approx g_{c}\sum_{l=-P}^{P+L_{h}-1}s\left[m-l\right]\;\mathrm{sinc}\left(l-\Delta -f_{s}\tau_{n}\right) \end{array}$$
where the summation is truncated around the mainlobe of the sinc function. The approximation in (Equation 5) holds for P∈N and $P-\Delta -f_{s}\tau_{n}\gg 1$. $L_{h}\in N$ is the length of the FIR filter to be defined later. We can formulate (Equation 5) as
(6)$$x_{n}\left[m\right]=g_{c}\;g_{n}^{T}\left(\theta_{d},d\right)\;s\left[m\right]$$
where s[m] contains $L=2P+L_{h}$ successive samples of the signal s[m]:(7)$$s\left[m\right]={\left[\begin{array}{cccc} s[m+P] & s[m+P-1] & \ldots & s[m-P-L_{h}+1] \end{array}\right]}^{T}$$
and $g_{n}\left(\theta_{d},d\right)$ is given by
(8)$$\begin{array}{cc} g_{n}\left(\theta_{d},d\right)= & \\ \; & \;{\left[\begin{array}{cccc} \mathrm{sinc}(-P-\Delta -f_{s}\tau_{n}) & \mathrm{sinc}(-P+1-\Delta -f_{s}\tau_{n}) & \ldots & \mathrm{sinc}(P+L_{h}-1-\Delta -f_{s}\tau_{n}) \end{array}\right]}^{T}\;. \end{array}$$

Considering $L_{h}$ successive time samples of the *n*th sensor signal, (Equation 6) becomes a vector of length $L_{h}$:(9)$$\begin{array}{cc} x_{n}\left[m\right] & ={\left[\begin{array}{cccc} x_{n}\left[m\right] & x_{n}\left[m-1\right] & \ldots & x_{n}\left[m-L_{h}+1\right] \end{array}\right]}^{T}\\ \; & =g_{c}\;G_{n}\left(\theta_{d},d\right)\;s\left[m\right] \end{array}$$
where $G_{n}\left(\theta_{d},d\right)$ is a Toeplitz matrix of size $L_{h}\times L$ with
(10)$$\begin{array}{cc} {\left[G_{n}\left(\theta_{d},d\right)\right]}_{ij} & =\mathrm{sinc}\left(-P-i+j-\Delta -f_{s}\tau_{n}\right) \end{array}$$
where $i=0,\ldots ,L_{h}-1$ and j=0,…,L−1. By combining the samples from the *N* sensors, we get a vector of length $NL_{h}$:(11)$$\begin{array}{cc} x[m] & ={\left[\begin{array}{ccc} x_{0}{\left[m\right]}^{T} & x_{1}{\left[m\right]}^{T} & x_{N-1}{\left[m\right]}^{T} \end{array}\right]}^{T}\\ \; & =G\left(\theta_{d},d\right)s\left[m\right] \end{array}$$
where $G(\theta_{d},d)$ is a matrix of size $NL_{h}\times L$:(12)$$G\left(\theta_{d},d\right)=\left[\begin{array}{c} G_{0}\left(\theta_{d},d\right)\\ G_{1}\left(\theta_{d},d\right)\\ \vdots \\ G_{N-1}\left(\theta_{d},d\right) \end{array}\right]\;.$$

The signal x[m] is affected by the array’s geometry, represented by $\tau_{n}$ in each element of $G(\theta_{d},d)$. This is similar to the structure observed in the frequency-domain model. Therefore, the steering vector is the counterpart to $G(\theta_{d},d)$ in the frequency domain. This shows a direct relationship between time-domain signal processing and its frequency-domain equivalent through the array geometry. In order to simplify the exposition, we assume that $G(\theta_{d},d)$ is known in advance since it is target-dependent. In practice, the matrix G can be obtained by iterative solutions such as maximum likelihood (ML) [30].

## 3. Feedback Beamforming

This section discusses the feedback-based beamformer (FB) architecture used for spatial signal processing. The FB architecture is similar to an IIR-like filter. It uses a feedback loop to retransmit the signal $s_{\mathrm{fb}}\left(t\right)$, which is synthesized from the weighted aggregation of sensor samples:$$\begin{array}{cc} s_{\mathrm{fb}}\left(t\right) & =\sum_{k=0}^{N-1}\alpha_{k}x_{k}\left(t-\tau_{\mathrm{pd}}-\tau_{n}\right). \end{array}$$

This feedback loop creates a dynamic spatial processing environment, making it a novel approach for spatial signal processing. The details of this architecture are discussed in [17]. The architecture combines data collected by sensors and processes it through two weighted sums called α and β. In the time domain, $L_{h}$ consecutive time samples are taken from each of the *N* sensors. α and β are vectors with dimensions of $NL_{h}$. These samples may contain desired signals and unwanted noise or interference from different directions. The system’s output is denoted as z(t) and is created using a weighted vector called β. The retransmitted signal combines the source signal and an additional weighted sum using the vector α. In order to demonstrate the presence of interference in this setup, Figure 2 introduces a noise source, $n_{t}$, placed at an angle $\theta_{n}$ relative to the array’s broadside. This configuration highlights the ability of the FB architecture to handle complicated signal environments by utilizing spatial feedback loops for improved signal processing.

By extending the signal model in (Equation 1) to the FB architecture and considering the noiseless case, the signal measured at the *n*th sensor is as follows [17]:(13)$$\begin{array}{cc} x_{n}\left(t\right) & =g_{c}\left(s\left(t-\tau_{\mathrm{pd}}-\tau_{n}\right)+s_{\mathrm{fb}}\left(t\right)\right)\\ \; & =g_{c}\left(s\left(t-\tau_{\mathrm{pd}}-\tau_{n}\right)+\sum_{k=0}^{N-1}\alpha_{k}x_{k}\left(t-\tau_{\mathrm{pd}}-\tau_{n}\right)\right). \end{array}$$

By using Shannon’s sampling theorem, as is the case in (Equation 5), the discrete-time signal can be written as
(14)$$x_{n}\left[m\right]\approx g_{c}\left[\sum_{l=-P}^{P+L_{h}-1}s\left[m-l\right]\mathrm{sinc}\left(l-\Delta -f_{s}\tau_{n}\right)+\sum_{l=-P}^{P+L_{h}-1}\sum_{k=0}^{N-1}\alpha_{k}^{T}x_{k}\left[m-l\right]\mathrm{sinc}\left(l-\Delta -f_{s}\tau_{n}\right)\right]\;$$
where $\alpha_{k}$ is a vector of length $L_{h}$ that defines the weights for the time samples of the *k*th sensor:$$\begin{array}{cc} \alpha_{k} & ={\left[\begin{array}{cccc} \alpha_{k}\left[0\right] & \alpha_{k}\left[1\right] & \ldots & \alpha_{k}\left[L_{h}-1\right] \end{array}\right]}^{T}. \end{array}$$

By simplifying the expression above, (Equation 14) can be rewritten as
(15)$$x_{n}\left[m\right]=g_{c}\left\{\sum_{l=-P}^{P+L_{h}-1}\left[\mathrm{sinc}\left(l-\Delta -f_{s}\tau_{n}\right)\;\left(s\left[m-l\right]+\sum_{k=0}^{N-1}\alpha_{k}^{T}x_{k}\left[m-l\right]\right)\right]\right\}.$$

Combining both (Equation 9) and (Equation 15) results in
(16)$$x_{n}\left[m\right]=g_{c}G_{n}\left(\theta_{d},d\right)\left[s\left[m\right]+i_{l}\alpha^{T}x\left[m\right]\right]$$
where $i_{l}$ is the *l*th column of the L×L identity matrix, $I_{L}$, and α is a vector of length $NL_{h}$, consisting of the weights for all the samples. By unifying all the sensors together, we obtain
(17)$$x\left[m\right]=g_{c}G\left(\theta_{d},d\right)\left[s\left[m\right]+i_{l}\alpha^{T}x\left[m\right]\right].$$

The above can be simplified to
(18)$$\begin{array}{c} \left[1-g_{c}\alpha^{T}G\left(\theta_{d},d\right)i_{l}\right]x\left[m\right]=g_{c}G\left(\theta_{d},d\right)s\left[m\right]. \end{array}$$

Thus, the relation between the input signal and the sensors’ samples is given by
(19)$$x\left[m\right]=\frac{g_{c}G\left(\theta_{d},d\right)s\left[m\right]}{1-g_{c}\alpha^{T}G\left(\theta_{d},d\right)i_{l}}.$$

In principle, any element of the input vector signal s[m] can be considered the desired signal. Moreover, based on the proposed feedback beamforming architecture, the output of the beamformer can be represented as $z\left[m\right]=\beta^{T}x\left[m\right]$. Therefore, the relationship between the input signal and FB output can be expressed as follows:(20)$$H_{\beta ,\alpha }\overset{\Delta }{=}\frac{z}{s}=\frac{g_{c}\beta^{T}G\left(\theta_{d},d\right)i_{l}}{1-g_{c}\alpha^{T}G\left(\theta_{d},d\right)i_{l}}\;.$$

After analyzing both the time-domain response and the frequency-domain response outlined in [17], it is apparent that they are closely related. The steering vector utilized in the frequency domain is similar to the construct $G\left(\theta_{d},d\right)i_{l}$ in the time domain. Additionally, the phase shift component $e^{-j\phi }$ in the frequency domain is comparable to fine-tuning the interpolation matrix $G(\theta_{d},d)$ in the time domain by adjusting the propagation delay associated with the target’s range estimation. Thus, this adjustment shows the alignment between time-domain processing and its frequency-domain counterpart by manipulating the coefficient interpolation matrix.

## 4. Finding the Optimal Weights

The optimal selection of the beamformer weights is a crucial factor in its performance. Unfortunately, even slight deviations from optimal weights may significantly impact beamformer efficacy, a concept that will be further explored in the subsequent sections. When compared to a conventional beamformer (CB), the unique aspect of the feedback beamformer is the introduction of spatial feedback through the retransmitted signal. This retransmission adds a layer of complexity and potential for enhanced performance by incorporating additional system information, as detailed in [17]. The efficacy of the FB hinges on the precise estimation of the DOA ($\theta_{d}$) and the target’s range (d). There are several ways to assess the estimations’ accuracy. One potential method of determining the provided feedback mechanism’s impact is quantifying the system’s supplementary information. To this end, the Cramér-Rao bound (CRB) was used, leveraging the Fisher information matrix (FIM). Given that the analysis and development of the beamformer’s response are conducted within the time domain, we utilized the time-domain FIM to evaluate the performance potential of the FB system.

The (m,n)th FIM element is given by [31]
(21)$$\begin{array}{c} {\left[J(\zeta )\right]}_{m,n}={\left|\frac{\partial z(\zeta )}{\partial \zeta_{m}}\right|}^{T}R^{-1}\left(\zeta \right)\left|\frac{\partial z(\zeta )}{\partial \zeta_{n}}\right|+\frac{1}{2}\mathrm{tr}\left[R^{-1}\left(\zeta \right)\frac{\partial R(\zeta )}{\partial \zeta_{m}}R^{-1}\left(\zeta \right)\frac{\partial R(\zeta )}{\partial \zeta_{n}}\right] \end{array}$$
where $\zeta =[\theta_{d},d]$ represents the vector of the parameters, and tr(·) denotes the trace operation on a matrix. The variables m,n∈1,2 specify the estimated parameters, and R(ζ) represents the N×N noise covariance matrix. By assuming the presence of white noise, which is statistically independent of the parameter vector ζ, and given that z(t) is scalar, expression (Equation 21) can be simplified as
(22)$${\left[J(\zeta )\right]}_{m,n}=\left|\frac{\partial z(\zeta )}{\partial \zeta_{m}}\right|\;\left|\frac{\partial z(\zeta )}{\partial \zeta_{n}}\right|.$$

The above derivatives yield the following main diagonal elements:(23)$$\begin{array}{cc} J_{\theta_{d}\theta_{d}} & =\frac{f_{1}}{{\left[1-g_{c}\alpha^{T}G\left(\theta_{d},d\right)i_{l}\right]}^{4}}\\ J_{dd} & =\frac{f_{2}}{{\left[1-g_{c}\alpha^{T}G\left(\theta_{d},d\right)i_{l}\right]}^{4}} \end{array}$$
where $f_{1}$ and $f_{2}$ are functions of the parameter set α, β, and $G(\theta_{d},d)$ and its partial derivatives. A detailed proof of (Equation 23) is given in the Appendix A. In order to obtain the optimal and precise estimate, it is required to maximize the diagonal elements of the FIM, $J_{\theta_{d}\theta_{d}}$ and $J_{dd}$. This maximization is carried out by minimizing the denominator $\left|1-g_{c}\alpha^{T}G\left(\theta_{d},d\right)i_{l}\right|$.

The optimal weight α can be written as
(24)$$\alpha_{\mathrm{opt}}^{T}=\frac{1}{{\hat{g}}_{c}}i_{l}^{T}G^{T}\left(\theta_{d},d\right){\left[G\left(\theta_{d},d\right)G^{T}\left(\theta_{d},d\right)\right]}^{-1}$$
where ${\hat{g}}_{c}$ is the channel gain estimate. For simplicity, the output weight is $\beta =\alpha_{\mathrm{opt}}$. The optimal beamformer weights vector contains the exact values for the DOA and the target range, which, in practice, are unknown. Thus, the optimal weights, considering both the DOA estimate ${\hat{\theta }}_{d}$ and the range estimate $\hat{d}$, are
(25)$$\alpha_{\mathrm{opt}}^{T}=\beta_{\mathrm{opt}}^{T}=\frac{1}{{\hat{g}}_{c}}i_{l}^{T}G^{T}\left({\hat{\theta }}_{d},\hat{d}\right){\left[G\left({\hat{\theta }}_{d},\hat{d}\right)G^{T}\left({\hat{\theta }}_{d},\hat{d}\right)\right]}^{-1}.$$

Substituting (Equation 25) into (Equation 20) yields the optimal beamformer time response:(26)$$\begin{array}{cc} H_{\beta_{\mathrm{opt}},\alpha_{\mathrm{opt}}} & =\frac{r\;i_{l}^{T}G^{T}\left({\hat{\theta }}_{d},\hat{d}\right){\left[G\left({\hat{\theta }}_{d},\hat{d}\right)G^{T}\left({\hat{\theta }}_{d},\hat{d}\right)\right]}^{-1}G\left(\theta_{d},d\right)i_{l}}{1-r\;i_{l}^{T}G^{T}\left({\hat{\theta }}_{d},\hat{d}\right){\left[G\left({\hat{\theta }}_{d},\hat{d}\right)G^{T}\left({\hat{\theta }}_{d},\hat{d}\right)\right]}^{-1}G\left(\theta_{d},d\right)i_{l}} \end{array}$$
where $r≜\frac{g_{c}}{{\hat{g}}_{c}}$ is the channel gain error (i.e., for accurate gain match r→1).

## 5. Beamformer Evaluation

In order to assess the beamformer’s efficacy, it is essential to consider a range of architecture-specific (such as *N* and δ) and target-specific (such as the DOA and the target’s distance from the array) parameters. This section examines the FB response, as detailed in (Equation 26), and evaluates its performance using key metrics such as sidelobe level reduction, directivity factor (DF), and beam width. These metrics are standard for assessing the performance of ULA beamformers. The examination was tested on the beam pattern of the proposed beamformer, which is defined by $10log_{10}{\left|H_{\beta_{\mathrm{opt}},\alpha_{\mathrm{opt}}}\right|}^{2}$ in units of dB. The FB’s performance metrics were benchmarked against conventional ULA beamformers, including the delay and sum (DS) and the maximum DF beamformers, which lack the FB’s feedback mechanism. This comparative analysis will highlight FB’s advancements and its potential impact.

### 5.1. Channel Gain Estimation Error

In the previous section, we defined the channel gain error as $r≜\frac{g_{c}}{{\hat{g}}_{c}}$. In the following subsection, we will investigate the influence of different estimations of *r* on the proposed beamformer response. Figure 3 demonstrates the beampattern for several channel gain estimation errors. As can be seen in Figure 3, *r* represents the array aperture. The closer the value is to 1 (precise gain alignment), the narrower the mainlobe, and the better the sidelobe attenuation. It is noteworthy that when achieving precise gain matching (i.e., as *r* tends towards 1), the beamformer achieves impeccable spatial selectivity.

### 5.2. Mainlobe and Sidelobe Attenuation

The accuracy of a beamformer is determined by the width of its center lobe and the height of its sidelobes. The behavior of a proposed feedback beamformer compared to the DS and maximum DF beamformers, which is presented in [29], is shown in Figure 4, depending on θ. To this end, we assume perfect range estimation, that is, $d=\hat{d}$. The DS beamformer is a basic beamformer obtained by the sum of the time-shifted versions of the sensor signals. The maximum DF beamformer tends to narrow the mainlobe and attenuate the sidelobes. As demonstrated in Figure 4, the feedback beamformer beampattern has a much narrower mainlobe than the DS beamformer. Additionally, the feedback beamformer has better attenuation (in the order of magnitude) than the DS beamformer. Furthermore, the FB mainlobe is almost as narrow as the maximum DF beamformer mainlobe. This indicates that the beamformer filter is spatially effective, nearly as much as a beamformer explicitly designed for that purpose. Moreover, while the distortionless maximum DF beamformer is designed to have a narrow beamwidth, it comes at the cost of high sidelobes. The feedback beamformer closely matches the mainlobe beamwidth while having much better sidelobe attenuation. These results indicate that our proposed beamformer can achieve better localization estimation for two of the most common beamformer performance measures.

### 5.3. Directivity Factor

The directivity factor (DF) is an important quality factor of the beamformer. It refers to the level of directionality of the beamformer. It is measured by the ratio between the beamformer gain in a specific direction and the average gain in all directions. In the time domain, it is defined as [29]
(27)$$D\left[\alpha ,\beta ,cos\left(\theta_{d}\right),d\right]=\frac{|B\left(\alpha ,\beta ,cos\left(\theta_{d}\right),d\right){|}^{2}}{\frac{1}{2}\int_{0}^{\pi }{\left|B\left(\alpha ,\beta ,cos\left(\theta \right),d\right)\right|}^{2}sin\left(\theta \right)\;d\theta }\;$$
where *B* represents the beampattern. As explained and developed in [32], the DF for a generic ULA is
(28)$$D=2\frac{N\delta }{\lambda }$$
for large values of *N* and λ≫δ.

Furthermore, as explained in detail in [17] and shown in Figure 3, the array aperture is a function of *r*. For r=0, the standard ULA beamformer is obtained, while for increasing values of *r*, the array’s aperture is increased. In order to avoid spatial ambiguities in the results of the directivity factor, r=0.6 was used throughout the entire experiment. By simulating the DF for both the feedback beamformer architecture and generic ULA and plotting the result as a function of δ, in Figure 5, we show that the DF was significantly improved. In addition, it can be seen from Figure 5 that when the number of sensors is increased, or the distance between sensors is increased, the DF of both cases increases as well due to better spatial resolution.

### 5.4. Range Error Influence

In CB, although the target’s DOA is the only important parameter, the range is estimated to obtain the target’s position more accurately in our design. We considered the perfect estimation of the target’s range for equal comparison between our architecture and CB. The assumption of a perfect estimation is carried out by setting $d=\hat{d}$ in (Equation 26), where $\hat{d}$ denotes the estimated distance from the reference sensor to the target and *d* signifies the target’s true range. Estimating the target’s range can increase the level of accuracy of a target to a specific location instead of a particular direction.

In order to check the influence of the target’s range estimation error, we further investigated the change in the beamformer response given in (Equation 26) when $d\neq \hat{d}$.

We have simulated the proposed beamformer beampattern with different range errors as a function of θ in Figure 6. It can be seen that the range estimation error can change beamformer performance by way of shifting the mainlobe direction and different attenuation on both sides of the mainlobe. As opposed to the range error influence in the frequency domain [17], the time-domain implementation is significantly less sensitive regarding the range error. This robustness can be seen in the way that the range estimation error negligibly affects the mainlobe position in the time domain. In contrast, small changes in the frequency domain can shift the position of the mainlobe by tens of degrees. In [17], this estimation error was addressed by adding another feedback beamformer. The minor effect of the estimation error in the time domain eliminates the use of an additional beamformer, which results in less hardware and a smaller footprint, which are critical considerations.

## 6. Time Domain vs. Frequency Domain Feedback Beamforming

In order to thoroughly investigate the proposed filter, we compared the proposed beamformer and the frequency domain beamformer [17]. Some of the standard comparison measures include beamformer fidelity, the system’s robustness, and computational complexity [33]. A comparison was made between time-frequency domain implementations according to the following additional parameters: the number of sensors used for the desired output, SNR, sampling frequency, data storage, processing requirements, and area considerations for implementing the hardware.

### 6.1. Calculation Complexity

Time-domain spatial filters offer distinct advantages over their frequency-domain counterparts, notably in computational efficiency. A significant benefit of time-domain processing, as demonstrated in the filter design outlined above (Equation 26), is its lower computational complexity relative to an analogous design in the frequency domain. Specifically, the most computationally intensive operation in the time-domain design is given by
$$G^{T}\left({\hat{\theta }}_{d},\hat{d}\right){\left[G\left({\hat{\theta }}_{d},\hat{d}\right)G^{T}\left({\hat{\theta }}_{d},\hat{d}\right)\right]}^{-1}G\left(\theta_{d},d\right).$$

This operation underscores the efficiency of time-domain designs in handling spatial filtering tasks. The focus is on minimizing the processing load without compromising the accuracy or effectiveness of the filtering process.

The above matrix multiplications require approximately $3N^{2}L_{h}^{3}$ multiplications and the same number of additions, where N≫1 and $L\approx L_{h}$. In the frequency-domain design, as defined and developed in [17], the beamformer is
(29)$$\begin{array}{cc} H & =\frac{g\beta^{H}de^{-j\phi }}{1-g\alpha^{H}de^{-j\phi }}\\ \; & ={\left[1-g\frac{e^{j\hat{\phi }}{\hat{d}}^{H}}{\hat{g}\left|\right|\hat{d}{\left|\right|}^{2}}de^{-j\phi }\right]}^{-1}g\frac{e^{j\hat{\phi }}{\hat{d}}^{H}}{\hat{g}\left|\right|\hat{d}{\left|\right|}^{2}}de^{-j\phi }\; \end{array}$$
where d is the steering vector, for which the *n*th element is $d_{n}=exp\left(-j\omega \tau_{n}\right)$, and ϕ is the round-trip signal propagation phase. One must consider the transform of each sensor signal to the frequency domain using the fast Fourier transform (FFT) and then use inverse FFT. Each time frame contains $L_{h}$ samples, so each sensor requires $L_{h}\log_{2}L_{h}$ operations. The FFT requires $NL_{h}\log_{2}L_{h}$ operations. The most expensive operation in the filter design (Equation 29) is the multiplication of the steering vector by itself, which requires $4NL_{h}$ multiplications and the same number of additions, where the factor of 4 is due to the frequency-domain design dealing with complex numbers. The frequency-domain design requires approximately $4N^{2}L_{h}^{2}\log_{2}L_{h}$ multiplications and a similar number of additions. This shows that for sufficiently large $NL_{h}$, the frequency-domain beamformer incurs lower computational costs. Certain applications do not prioritize the beamformer’s output signal in the time domain, so utilizing the frequency-domain beamformer would be more efficient. The above frequency domain’s complexity calculation assumes that $L_{h}$ is an integer power of two. Otherwise, due to the FFT algorithm, zero padding must be added to each frame, which increases the computational complexity even more.

### 6.2. Execution Time

Radar processing systems are based on pulse signals rather than continuous-wave (CW) stimuli. Although CW signals simplify system analysis, they are hardly feasible to implement and waste a significant part of the system’s power; therefore, they are barely used. Moreover, CW radar systems rarely measure distance, which is our primary goal. One main drawback when using pulse-based signals is the known fact that pulse signals, which are finite in the time domain, contain high frequencies. A practical pulse signal contains many more frequencies than a CW signal, which includes, in theory, only one frequency. Each of the frequencies is considered in the steering vector in the frequency domain, increasing the execution time of such a beamformer. Figure 7 demonstrates the differences in the time domain and in the power spectrum between CW and a pulse signal. As seen in the figures, the frequencies that must be considered are orders of magnitudes larger in the pulse-based signals than in the sine wave signals. Although just the sine frequency is a reasonable power in a sine wave, for the pulse-based signal, the power remains quite similar with increasing frequency.

This fact directly results from the execution time of the frequency domain beamformer implementation. In [17], the beamformer computational complexity remains the same with the number of sensors. In order to authenticate this claim, in Figure 8, we show the execution time of the feedback beamformer in both the time and frequency domains as a function of the array size. As can be seen from Figure 8, for a low number of sensors (up to 10 sensors), the time domain has a better execution time. In addition, one can see that the frequency domain calculation time is on the scale of seconds, which is not feasible in real-time systems. The exponential increase in calculation time in the time domain is due to the growth of (Equation 12), and the samples must be taken at each interval.

## 7. Conclusions

We have introduced a closed-form solution for a time-domain feedback-based beamformer, laying the groundwork for its future application and development. The feedback architecture demonstrated notable enhancements over traditional beamforming techniques, such as “delay and sum” and maximum directivity factor, for beamformers while also confirming its equivalence to frequency-domain approaches. We conducted an in-depth analysis of how range estimation errors influence the performance of the beamformer, revealing that the time-domain implementation offers superior robustness to these errors compared to its frequency-domain counterpart. The comparative study on computational complexity and execution time between time- and frequency-domain beamformers indicates that the choice between these implementations depends on specific use-case scenarios, particularly the number of sensors and the volume of sampled data. Notably, our time-domain beamformer exhibits improved execution times with a limited number of sensors, especially in applications utilizing the pulse-based signals that are standard in radar systems. This feature is crucial for real-time applications where execution speed is critical.

Future research directions for the time-domain FB will explore its application to ultra-wideband (UWB) signals, examining how pulse signals influence the results presented in this study, and identifying specific conditions under which these effects are most pronounced. Another area of interest involves extending the current findings to scenarios involving nonstationary or multiple targets, enhancing the beamformer’s applicability to dynamic environments. Additionally, the proposed implementation holds potential for broader applications across various spatial array processing tasks, suggesting its versatility in the field. A particularly promising avenue involves refining the design of feedback beamforming to enhance adaptive beamforming techniques. This includes dynamically adjusting the α and β coefficients within iterative processes to achieve improved performance compared to the static beamforming approach detailed in this research.

## Figures and Tables

**Figure 1 sensors-24-02179-f001:**
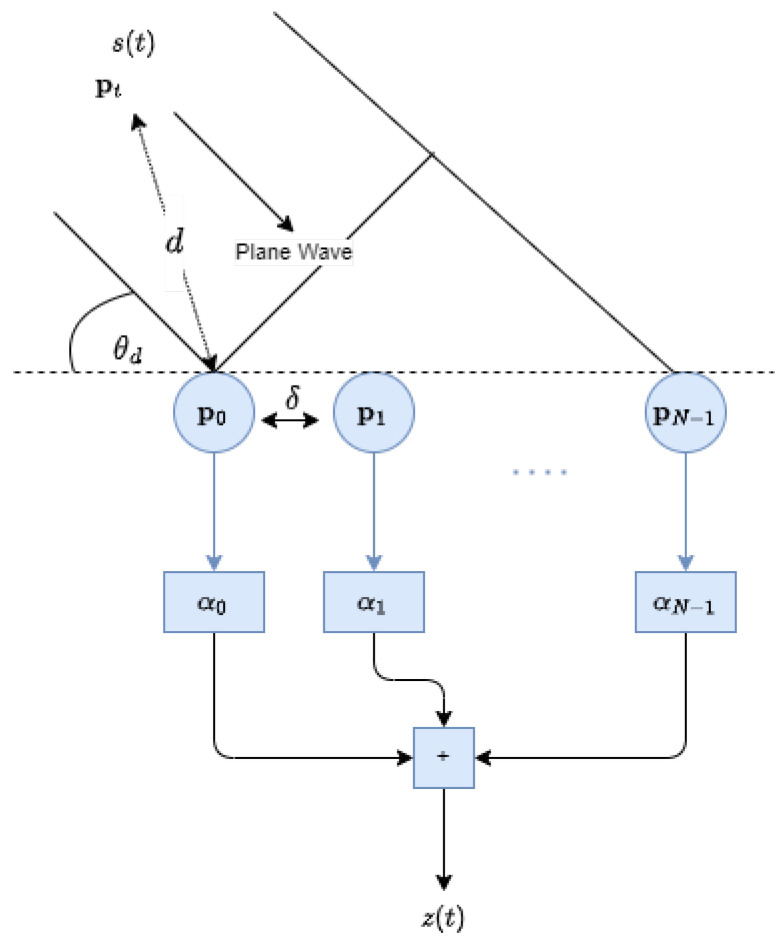
Uniform linear array.

**Figure 2 sensors-24-02179-f002:**
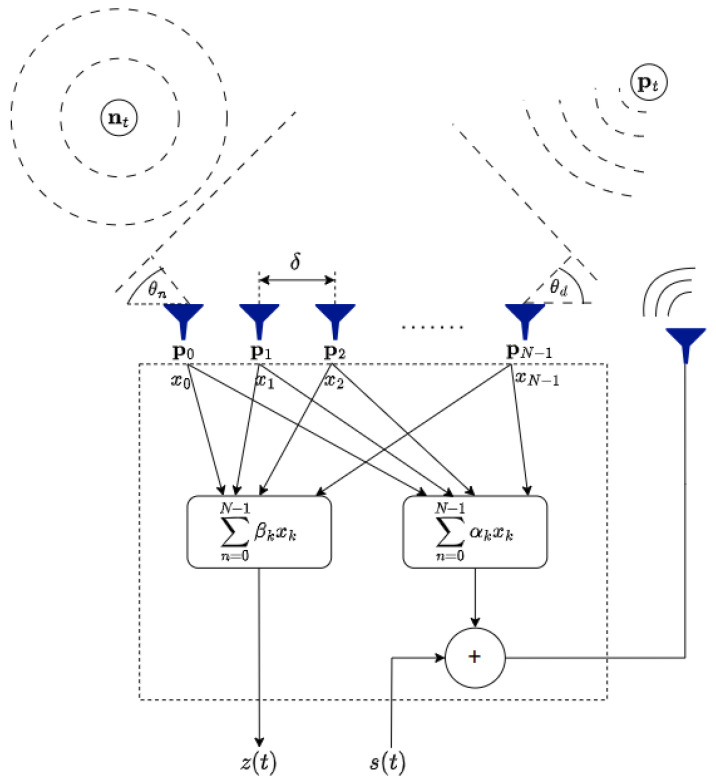
Proposed beamformer structure. The goal is to spatially localize the target, $P_{t}$, by retransmitting the signal.

**Figure 3 sensors-24-02179-f003:**
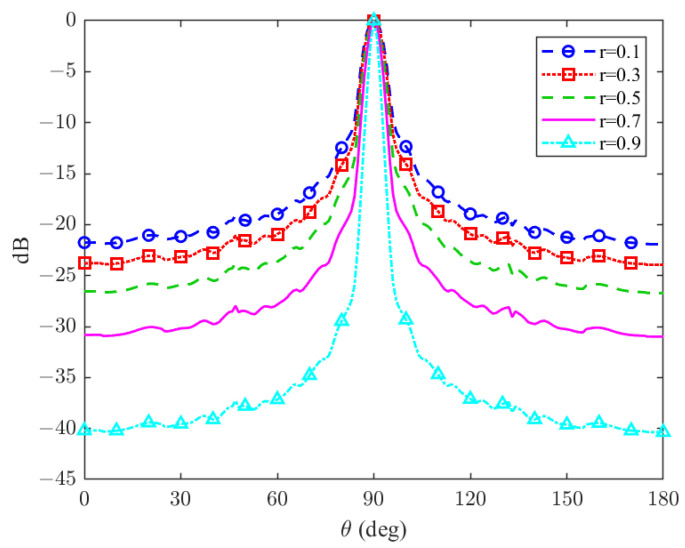
Feedback beamformer beampattern for different channel gain estimations errors for $\theta_{d}=90^{\circ }$ and N=10.

**Figure 4 sensors-24-02179-f004:**
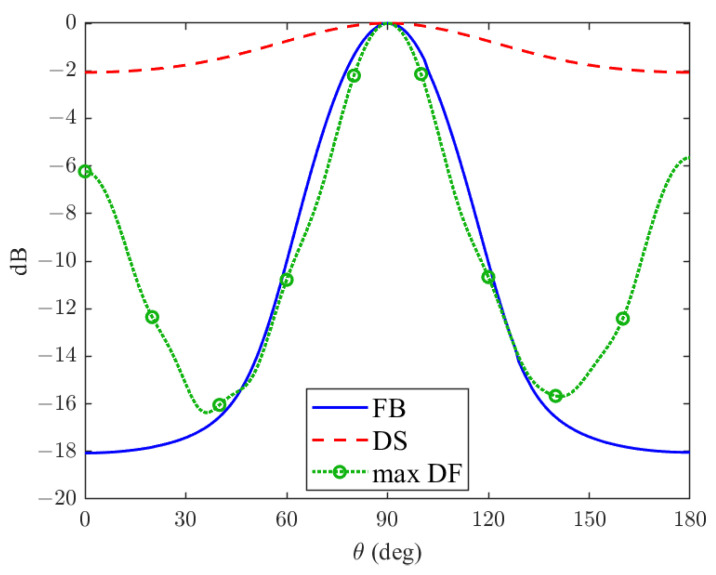
Feedback beamformer beampattern (blue line), compared to known ULA beamformers, DS (dashed red line), and maximum DF (dashed green line) for $\theta_{d}=90^{\circ }$ and r=0.6, respectively.

**Figure 5 sensors-24-02179-f005:**
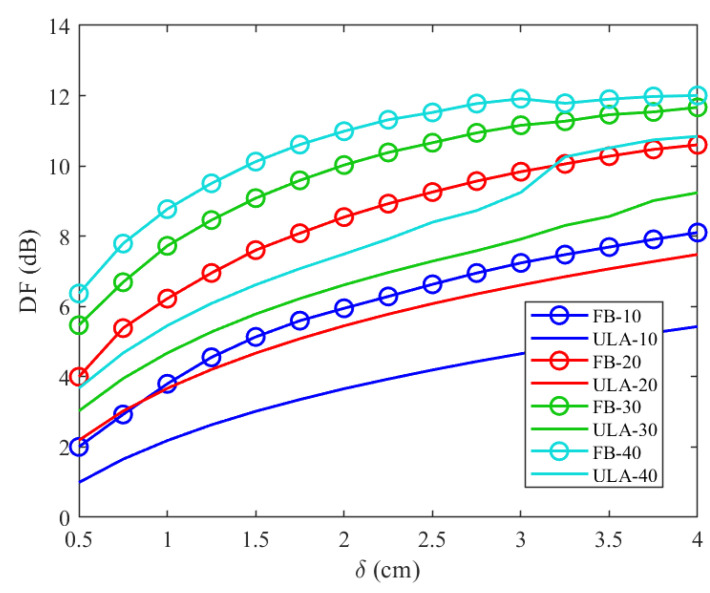
Directivity factor as a function of δ for the feedback beamformer (Equation 26) (lines with circles) compared to DF for a generic ULA (solid lines) for a signal wave with a frequency carrier of 1 GHz, r=0.6, $\theta_{d}=45^{\circ }$. DF was tested for several values of N: N=10 (blue lines), N=20 (red lines), N=30 (green lines), and N=40 (light blue lines).

**Figure 6 sensors-24-02179-f006:**
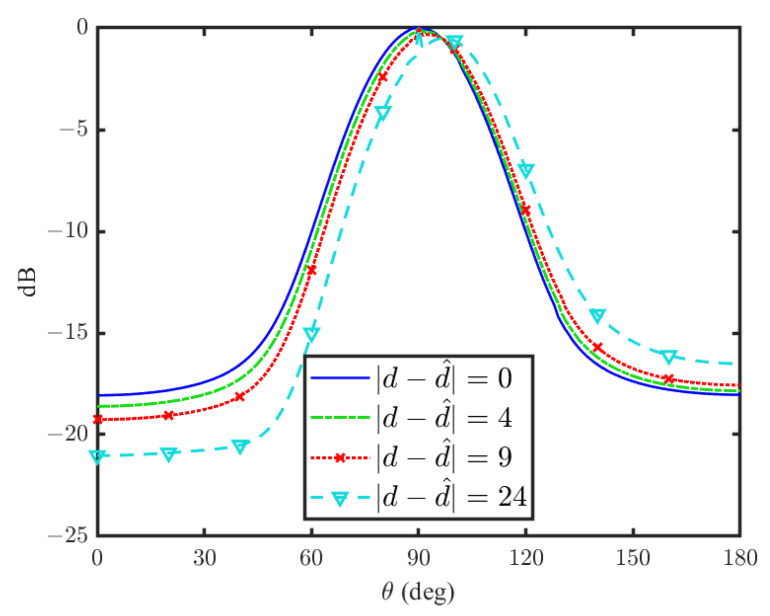
Effect of incorrect target range estimation for $\theta_{d}=90^{\circ }$ and r=0.6.

**Figure 7 sensors-24-02179-f007:**
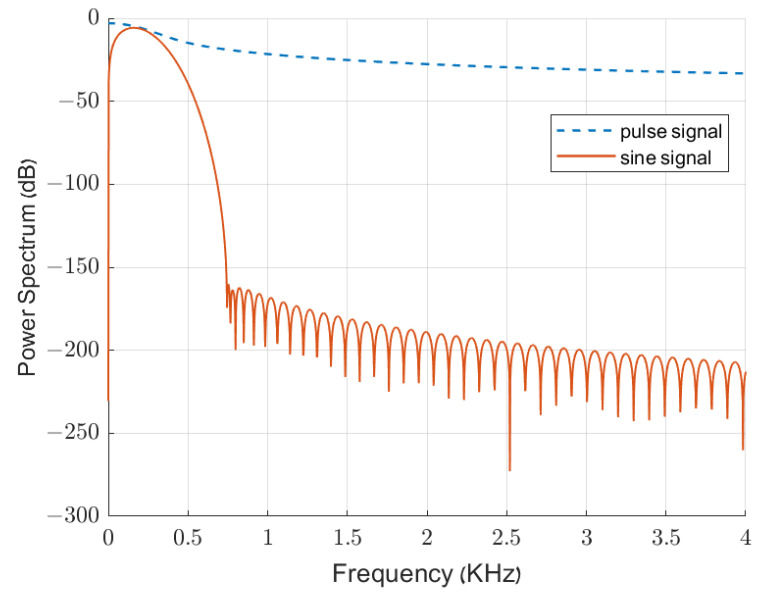
Power spectrum of both the pulse wave (blue line) and sine wave (red line) in the same sampling frequency.

**Figure 8 sensors-24-02179-f008:**
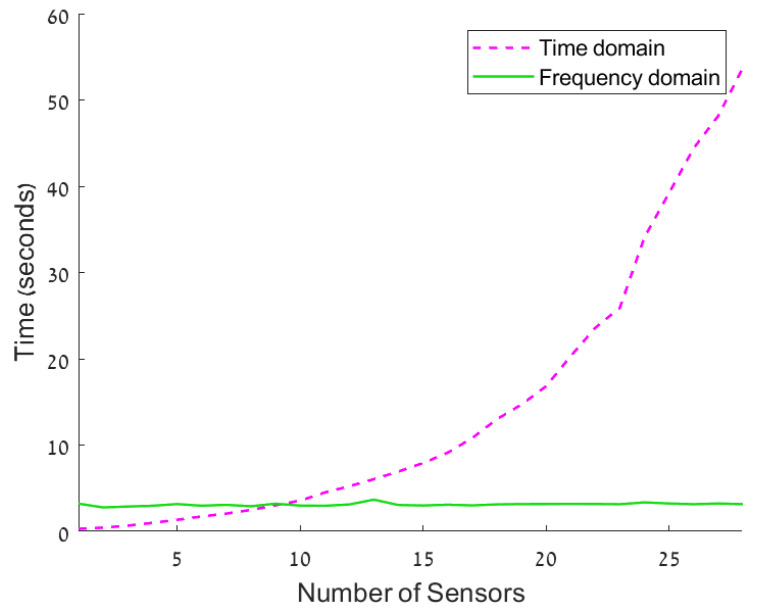
Execution time of the beamformer output for both time and frequency domains when 10 K frequencies are considered.

## Data Availability

Data are contained within the article.

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
