# Peer review of "Feedback Beamforming in the Time Domain"

_sensors, 2024, doi:10.3390/s24072179_

Round 1
Reviewer 1 Report
Comments and Suggestions for Authors
Please see the attached file.

Reviewer 2 Report
Comments and Suggestions for Authors
I see lot of fundamental loopholes in this paper.
1. What will be the units of fstn? Will it be same as t? If not, how is equation 1 valid.
2. What is (t-l) in equation 3? l is an integer, whereas t is time. It will a unit mismatch.
3. What is 'P' in equation 3? What is the basis for selecting the value for 'P'?
4. It is given that r = gc /gb as channel gain error. Is this the correct definition for error?
5. The integral in equation 25 has to be rechecked. The term in the numerator will get cancelled with similar term in denominator, which are independent of θ.
6. In figure 4, it is shown that directivity increases with antenna spacing. This may be due to increase in aperture, but it will create spatial ambiguities. The experiment should be for the same aperture with varying element spacing.
7. I do not see any algorithm for estimating the range d.
Reviewer 3 Report
Comments and Suggestions for Authors
* Fig.1 - consider drawing an additional line, which pinpoints a distance between $p_t$ and $p_0$. Right now it is not clear how you measure it: a) from $p_t$ to the "plain wave" line, or b) as normal geometrical distance
* pg.3/l.96 - the length (in the time domain) of the rectangular function should be provided
* pg.3/l.100 - $N$ shouldn't be $n$ ?
* eq (11)&(12) - please explain why the bound of the sum has been changed comparing eq (3)
* eq (15) - it should be as follows:
$$ [1-g_c \mathbf i_l \mathbf \alpha^T \mathbf G(\theta_d,d)]\mathbf x(t) = ...$$
* pg.6/l.159 - a lack of bracket before $\theta_d$
* pg.7/l.184 - shouldn't be $N$ instead of $M$ ?
* pg.8/l.216 - whole subsection 5.3 and Fig 5 discuss the influence of range error, but there is a lack of definition of this error; should be provided (similar to the subsection touching directivity factor) to better understand what you have in mind
* what for Fig.6 and Fig. 7 are presented? It is obvious that the pulse has a very broad spectrum. But in radar technique, they use shaped pulse, which allows controlling frequency band used by the emitted signal.
* comparing computational costs - pg.10/l.246 and pg.10/l.256 one can see that for large enough $L_h$ the frequency-beamformer is less computationally expensive; taking into consideration that in some applications we don't care about the beamformer's output signal in time domain it would be much efficient to use frequency-beamformer - please comment this fact
General question - there is no solution to finding the DOA, consequently, a question is how to obtain the matrix $\mamth G(\theta_d,d)$? How is the angle inferred?
Reviewer 4 Report
Comments and Suggestions for Authors
Please see the attached file.
1. There are some writing and formatting errors in the paper, like
Line 115, the symbol ‘S ’ doesn’t appear in the subsequent formulas
Line 169 equation (17) should be equation (18)
2. Although some descriptions are showed in the figure’s title, it is still hard to understand the results in the figures. Authors should make the figures more clear and add legends to these figures.
3. In line 176, a channel gain estimate is mentioned and utilized for the proposed method, authors should point out how to get or calculate in the framework and whether other methods also need this estimate.
4. Author should give the reference number for the compared methods .For example, in section 5.1, they compare the method with ULA beamformer and maximum DF beamformer but the reference numbers are not listed. Meanwhile, the discussion about the experiment results should be more detailed.

Round 2
Reviewer 1 Report
Comments and Suggestions for Authors
The authors have well addressed all my concerns, no further comments.
Reviewer 2 Report
Comments and Suggestions for Authors
I am still concerned with the variables used in the equations. All the variables in the argument of equation 1 should be same. Obviously, the delay is in seconds, thus the variable 't' should be in seconds. However, the third variable in that equation is just the samples( number). If sampling has to be done, it should also be applied to time 't' and delay variable.
Once this aspect is taken care, the other equations will follow the suit.
